# PBADET: A ONE-STAGE ANCHOR-FREE APPROACH FOR PART-BODY ASSOCIATION

**Zhongpai Gao**[1]**, Huayi Zhou**[2]**, Abhishek Sharma**[1]**, Meng Zheng**[1]**, Benjamin Planche**[1]**,
Terrence Chen**[1]**, Ziyan Wu**[1]

[1]United Imaging Intelligence, Burlington, MA 01803, USA
[2]Shanghai Jiao Tong University, Shanghai 200240, China
{first.last}@uii-ai.com, sjtu_zhy@sjtu.edu.cn

## ABSTRACT

The detection of human parts (*e.g.*, hands, face) and their correct association with individuals is an essential task, *e.g.*, for ubiquitous human-machine interfaces and action recognition. Traditional methods often employ multi-stage processes, rely on cumbersome anchor-based systems, or do not scale well to larger part sets. This paper presents *PBADet*, a novel one-stage, anchor-free approach for part-body association detection. Building upon the anchor-free object representation across multi-scale feature maps, we introduce a singular part-to-body center offset that effectively encapsulates the relationship between parts and their parent bodies. Our design is inherently versatile and capable of managing multiple parts-to-body associations without compromising on detection accuracy or robustness. Comprehensive experiments on various datasets underscore the efficacy of our approach, which not only outperforms existing state-of-the-art techniques but also offers a more streamlined and efficient solution to the part-body association challenge.

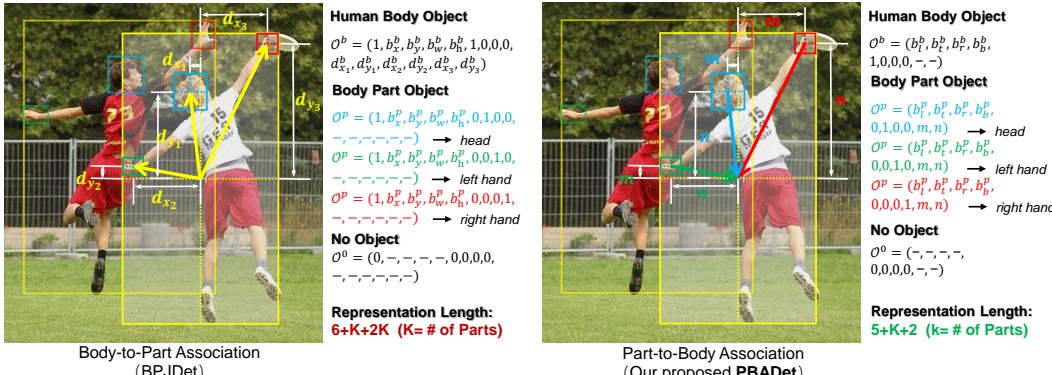

Figure 1: A comparative illustration between BPJDet (Zhou et al., 2023a) and our *PBADet* in terms of the extended representations. Using the joint detection of three parts—head, left hand, and right hand—as an example, the original BPJDet's body-to-part configuration demands an extended representation of length 15 (6+K+2K, K=3). Due to unused positions in the part object representation, this approach can result in inefficiencies and redundancies. In contrast, our *PBADet*, operating on a part-to-body principle, adopts a more concise representation of length 10 (5+K+2, K=3), optimizing for greater utilization and efficiency.

## 1 INTRODUCTION

Human part-body association refers to the task of detecting human parts within an image and identifying the location of the corresponding person for each detected part, *e.g.*, left and right hands, face, left and right feet, *etc*. This task is vital in scenarios where multiple individuals are present, and, *e.g.*, specific gestures from a particular person must be recognized and acted upon. An illustrative example is found in medical scan rooms, where technicians use hand gestures to locate the scanning area of a patient and initiate the scanning process. In such intricate scenarios, it is crucial that the

system responds only to the technician's gestures, effectively filtering out the patient's hands to avoid unintended interference. By achieving this nuanced recognition, part-body association serves as a foundational technology across various fields, including human-computer interaction, virtual reality, robotics, and medical analysis, paving the way for more intuitive and precise control systems.

Despite the evident necessity of part-body association, achieving accuracy in complex scenarios remains a significant challenge. Existing approaches often involve two-stage processes that rely on heuristic strategies or learned association networks. For example, Zhou et al. (2018) presents an approach that detects hands and the human body pose, employing a heuristic matching strategy for hand-body association; specifically, they verify whether a left or right arm joint falls within a bounding box of a raised hand. BodyHands (Narasimhaswamy et al., 2022) proposes to detect hands and bodies separately, utilizing a hand-body association network to predict association scores between them. However, these methods predict hand-body relationships in two distinct stages. The two-stage nature potentially introduces complexity and possibly compromises efficiency, highlighting the need for a more streamlined approach to hand-body association.

BPJDet (Zhou et al., 2023b) introduces a body-part joint detector that can be integrated with one-stage anchor-based detectors, such as the YOLO series (Redmon et al., 2016; Redmon & Farhadi, 2018; Jocher et al., 2022; Wang et al., 2023b). This method extends object detection by incorporating center offsets from a body to specific body parts, such as the left and right hands, left and right feet, and head. These center offsets act as bridges, enabling body-part association with post-processing. However, the method poses some inherent challenges. Incorporating center offsets to each body part means that as the number of body parts increases, so does the complexity of the object detection representation, which causes the lack of flexibility to handle various numbers of body parts. This increase in complexity leads to a corresponding degradation in the overall detection performance. Furthermore, not all body parts are always visible within a body image, and when they are obscured, the corresponding center offsets become invalid, rendering the method less effective.

This paper introduces a novel one-stage anchor-free approach for part-body association, called *PBADet*, which creatively extends anchor-free object detection by introducing a single part-to-body center offset. The key innovations of our method are detailed in three primary aspects. First, anchor-free object detection methods (Tian et al., 2019; Feng et al., 2021) inherently contain information about part-body association, as each location within a bounding box predicts a 4D vector representing distances to the bounding box's sides. Our approach leverages this natural relationship by supplementing it with a 2D vector, denoting the distances from the location to the center of the body bounding box, thereby explicitly representing the part-body association in a logical extension to current techniques. Second, using a single part-body center offset, our method can accommodate any number of body parts without increasing the number of center offsets correspondingly. This scalable design avoids degrading the overall object detection performance and provides an elegant solution for part-body association. Lastly, unlike body-part center offsets, which may involve one-to-many correspondence and become invalid when some parts are invisible, our method guarantees a one-to-one correspondence between each part and a body. The part-body center offset is always valid, providing a well-defined ground truth for supervised training. In addition, this one-to-one correspondence simplifies post-processing and ensures precise part-body associations.

In summary, the contributions of this paper are as follows:

- We introduce a novel extension to one-stage anchor-free object detection methods for part-body association. This extension is characterized by incorporating a single part-body center offset, offering a streamlined and intuitive approach to identifying part-body relationships.

- Our proposed method serves as a universal framework, capable of supporting multiple parts-to-body associations. The innovative design ensures scalability without compromising the accuracy and robustness of the detection process.

- Through experiments, we validate that our method delivers state-of-the-art performance, demonstrating its efficacy and potential as a new standard in part-body association techniques.

## 2 RELATED WORK

**Body-part Joint Detection.** Human body and part detection fall within the broader category of general object detection (Girshick, 2015; Redmon et al., 2016; Liu et al., 2016) and have seen signif-

icant exploration and advancement in recent years (Dollar et al., 2011; Liu et al., 2019; Deng et al., 2020; Bambach et al., 2015). This surge in research has been facilitated by the availability of public human body and part datasets, such as COCO-WholeBody (Jin et al., 2020), COCO-HumanParts (Yang et al., 2020), CrowdHuman (Shao et al., 2018), Wider Face (Yang et al., 2016), among others. In the context of this rich landscape, our paper specifically targets the challenge of human body-part joint detection and association.

Previous works have approached the task of body-part joint detection using various strategies. DA-RNN (Zhang et al., 2019) proposes to detect bodies and heads in pairs using a double-anchor RPN to enhance person detection in crowded environments. JointDet (Chi et al., 2020b) incorporates a head-body relationship discriminating module to facilitate relational learning between human bodies and heads, aiming to improve human detection accuracy and reduce head false positives. BFJDet (Wan et al., 2021) introduces a head hook center in object representation and employs an embedding matching loss to associate the body and face of the same person. Hier R-CNN (Yang et al., 2020) extends the Mask R-CNN (He et al., 2017) pipeline, adopting a two-branch system to detect human bodies through the faster branch and human parts through an anchor-free Hier branch utilizing a per-pixel prediction mechanism. Detector-in-Detector (Li et al., 2019) takes a similar approach, using Faster R-CNN (Ren et al., 2015) with two detectors to separately focus on the human body and parts in a coarse-to-fine manner. Distinct from these methods, our paper introduces a one-stage anchor-free approach for hand-body association.

**Human-object Interaction.** Human-object interaction (HOI) and body-part association aim to understand spatial relationships but present distinct characteristics. In the realm of HOI, various methods have been proposed to handle different aspects of the problem. GPNN (Wang et al., 2020) treats HOI as a keypoint detection and grouping challenge, whereas HOTR (Kim et al., 2021) directly predicts human, object, and interaction triplets from an image, utilizing a transformer encoder-decoder architecture. Additionally, in the specialized field of hand-contact detection, Shan *et al*. (Shan et al., 2020) developed a hand-object detector that furnishes detailed hand attributes such as the location, side, contact state, and bounding box of the interacting object. ContactHands (Narasimhaswamy et al., 2020) builds upon this by localizing hands and deploying another object detector to determine physical contact. However, part-body association focuses solely on the connection between human parts and the body, simplifying the task but requiring specific attention to this nuanced relationship.

**Multi-person Pose Estimation.** Multi-person pose estimation, exemplified by works such as Jin et al. (2022); Nie et al. (2019); Newell et al. (2017), bears a significant resemblance to the task of hand-body association. . In Jin et al. (2022), the concept of centripetal offsets is introduced to effectively group human joints, thereby facilitating efficient and accurate multi-person pose estimation. YOLO-Pose (Maji et al., 2022), an advancement in the YOLO series, integrates bounding box detection with simultaneous 2D pose prediction for multiple individuals within a single processing framework. Similarly, ED-Pose (Yang et al., 2023) employs a dual explicit box detection approach to synergize human-level and keypoint-level contextual learning. Despite these advancements, multi-person pose estimation methods typically do not provide specific bounding boxes for human parts, an essential element for tasks like hand-body association.

Note that while Jin et al. (2022) and our proposed method both utilize part-to-body center offsets, their functionalities and roles within each framework are different. Jin et al. (2022) leverages these offsets primarily for grouping in pose estimation, whereas our method focuses on establishing a one-to-one correspondence for accurate part-body association.

**Anchor-free Detection.** The utilization of anchor boxes, featured prominently in detection architectures such as Faster R-CNN (Ren et al., 2015), SSD (Liu et al., 2016), and YOLOv3 (Redmon & Farhadi, 2018), has become foundational in many detection frameworks. Nevertheless, these anchor boxes introduce numerous hyper-parameters, necessitating meticulous tuning for optimal performance. Furthermore, they present challenges from the imbalance between positive and negative training samples. Crucially, for our specific task of part-body association, the part-body center offset can fall far outside the predefined anchor boxes designated for the parts. This characteristic makes the task of learning part-body association using anchor-based methods particularly challenging.

Recent years have witnessed the emergence of several anchor-free methods. YOLOv1 (Redmon et al., 2016), for example, determines bounding boxes using points proximal to object centers. How-

ever, this approach exhibits a lower recall than its anchor-based counterparts, primarily because it relies solely on near-center points. In a departure from this strategy, FCOS (Tian et al., 2019) leverages all points within a ground-truth bounding box, incorporating a centerness branch to suppress suboptimal detected boxes. However, this method performs object classification and localization independently, which may cause a lack of interaction between the two tasks, leading to inconsistent predictions. Instead of using a centerness branch, TOOD (Feng et al., 2021) introduces task alignment learning to enhance interaction between the two tasks for high-quality predictions. This refined concept from TOOD serves as the foundation for our part-body association method.

## 3 METHODOLOGY

### 3.1 TASK FORMULATION

Given an input image, we consider the task of regressing a set of bounding boxes, the corresponding classes (*i.e.*, body and each part), and the corresponding association offsets to the bodies.

**Anchor-free Dense Bounding-box Prediction.** Let $\boldsymbol{F}_i \in \mathbb{R}^{H_i \times W_i \times C_i}$ represents the feature map at layer $i \in \{1, \cdots, L\}$ of a backbone CNN, with $s_i$ being the total stride up to that layer; $H_i$, $W_i$, and $C_i$ the height, width, and depth of the feature maps respectively; and $L$ the number of layers whose feature maps are considered. We task a sub-network to regress the bounding-box of each target and another to classify the boxes, *i.e.*, to match the ground-truth targets $T = (B, c)$. Here $B = \{x_l, y_t, x_r, y_b\} \in \mathbb{R}^4$ denotes the coordinates of the left-top and right-bottom corners of the bounding box; $c \in \{1, 2, \cdots, N\}$ specifies the class of the object within the bounding box; $N$ stands for the total number of classes. For example, $N = 4$ when focusing on the body, left hand, right hand, and face. In the anchor-free paradigm, detection is formulated as a dense inference, *i.e.*, in a per-pixel prediction fashion in feature maps. For each position $p_i = (x_i, y_i)$ in $F_i$, the detection head regresses a 4D vector $(l_i, t_i, r_i, b_i)$, which represents the relative offsets from the four sides of a bounding box anchored in $p_i$. Based on the relation $(x_i, y_i) = (\lfloor \frac{x}{s_i} \rfloor, \lfloor \frac{y}{s_i} \rfloor)$ between feature map locations $p_i$ and corresponding locations $p = (x, y)$ in the original image, the predicted values should satisfy the following equations w.r.t. the ground-truth $B$:

$$\lfloor \frac{x_l}{s_i} \rfloor = x_i - l_i, \quad \lfloor \frac{y_t}{s_i} \rfloor = y_i - t_i, \quad \lfloor \frac{x_r}{s_i} \rfloor = x_i + r_i, \quad \lfloor \frac{y_b}{s_i} \rfloor = y_i + b_i. \quad (1)$$

Note that anchor points $p_i$ are selected from multi-level feature maps, which aids in detecting objects of varying sizes and enhances the robustness of predictions. Similarly, a classification head densely returns a score vector $o_c \in \mathbb{R}^N$ w.r.t. each class for each position in the feature maps.

**Part-to-body Association.** In anchor-free approaches, all points enclosed within a ground-truth bounding box are treated as *positive samples*, *i.e.*, only those are used to supervise the bounding box during training (Tian et al., 2019). We utilize this property to naturally enable part-to-body associativity: any anchor point belonging to a part's bounding-box should also belong to the body's bounding box; and thus should satisfy Equation (1) for both sets of bounding parameters. Hence, we can extend the part detection task to not only predicting the 4D vector corresponding to the part's own bounding box, but also a second vector pertaining to the corresponding body's bounding box. For a more concise delineation of the part-body association, we define the second vector as only the 2D center offset from part to body (*i.e.*, transitioning from a 4D vector to a 2D vector).

Therefore, we extend the ground-truth target as $T = \{B, c, c^b\} \in \mathbb{R}^4 \times \{1, 2, \cdots, N\} \times \mathbb{R}^2$, where $c^b = \{c_x^b, c_y^b\}$ is the center of the body that encloses the part. For an anchor point within the part's bounding box, we thus additionally regress a 2D vector $(m_i, n_i)$, representing the offset from the anchor point to the body center to encode the part-body association, which should satisfy:

$$\lfloor \frac{c_x^b}{s_i} \rfloor = x_i + \lambda m_i, \quad \lfloor \frac{c_y^b}{s_i} \rfloor = y_i + \lambda n_i, \quad (2)$$

where $\lambda$ is a scaling factor of $m_i$ and $n_i$ to control the range of the network outputs. The part-body association prediction is also performed over the multi-level feature maps.

In summary, our per-position network predictions can be denoted as $o = \{o_b, o_c, o_d\}$, where $o_b = \{l_i, t_i, r_i, b_i\}$ is the bounding box prediction, $o_c = \{c_1, \cdots, c_N\}$ is the classification result, and $o_d = \{m_i, n_i\}$ relates to the part-body association. It is worth noting that our discussion of the human

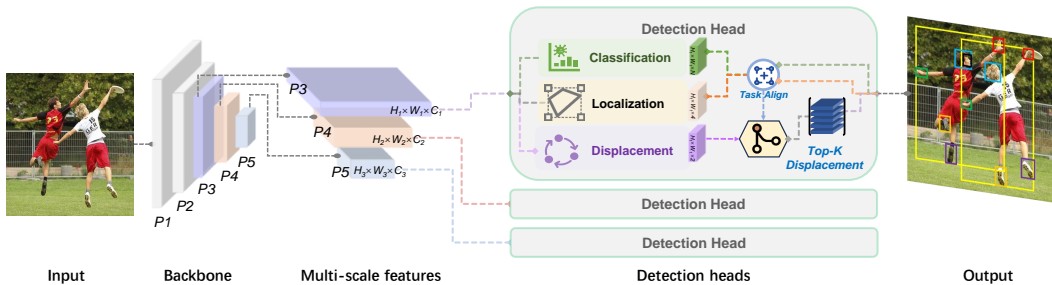

Figure 2: Illustration of the proposed pipeline.

part-body association serves as a representative example. However, this formulation stands as a universal framework, aptly addressing various parts-to-body association challenges (*e.g.*, the wheel-and-car association) without requiring any modifications. Since we define the part-body association as a 2D vector that denotes the center offset from the part to the body, it can seamlessly accommodate associations involving multiple parts.

## 3.2 NETWORK ARCHITECTURE

Our approach is founded on one-stage anchor-free detectors, including models like YOLOv5 (Jocher, 2020), YOLOv7 (Wang et al., 2023a), YOLOv8 (Jocher et al., 2023), among others. While YOLOv5 and YOLOv7 were originally designed as anchor-based methods, they can be conveniently adapted with anchor-free heads. Given the feature maps $\boldsymbol{F}_i \in \mathbb{R}^{H_i \times W_i \times C_i}$ produced by the backbone network as inputs, the detection head comprises three distinct output branches. These branches are responsible for bounding box prediction, class prediction, and part-body association prediction, respectively. Each branch is constructed using a three-layer convolutional network, where the kernel sizes are $\{3 \times 3, 3 \times 3, 1 \times 1\}$ and the stride is 1. Specifically, the bounding-box sub-network has the channel structure $\{C_i, \lfloor \frac{C_i}{4} \rfloor, \lfloor \frac{C_i}{4} \rfloor, 64\}$ and is followed by a DFL module (Li et al., 2022) to output $\boldsymbol{O}_b = \in \mathbb{R}^{H_i \times W_i \times 4}$ (2D map of $o_b$ predictions), the class branch adopts $\{C_i, C_i, C_i, N\}$ to output $\boldsymbol{O}_c \in \mathbb{R}^{H_i \times W_i \times N}$, and the part-body association branch uses $\{C_i, \lfloor \frac{C_i}{4} \rfloor, \lfloor \frac{C_i}{4} \rfloor, 2\}$ to output $\boldsymbol{O}_d \in \mathbb{R}^{H_i \times W_i \times 2}$.

## 3.3 LOSS FUNCTIONS

Our method incorporates the task-alignment learning strategy from TOOD (Feng et al., 2021) to supervise bounding box and class predictions, which includes a bounding-box IoU loss $\mathcal{L}_{iou}$, a bounding-box DFL loss $\mathcal{L}_{dfl}$, and a classification loss $\mathcal{L}_{cls}$ (refer to Feng et al. (2021) and Li et al. (2022) for the explicit definition of these functions). Furthermore, a dedicated part-body association loss is introduced for our specific objective.

Given the feature maps $\boldsymbol{F}_i \in \mathbb{R}^{H_i \times W_i \times C_i}$, the part-body association detection branch produces an output $\boldsymbol{O}_d \in \mathbb{R}^{H_i \times W_i \times 2}$, indicating that each anchor point yields a 2D vector. Drawing inspiration from the task-alignment learning concept, the anchor assignment for the part-body association remains consistent with those used for bounding-box and class supervision—"a well-aligned anchor point should be able to predict a high classification score with a precise localization jointly" (Feng et al., 2021). The anchor alignment metric is expressed as $t = s^\alpha \cdot u^\beta$, where $s$ and $u$ denote a classification score and an IoU value, respectively. $\alpha$ and $\beta$ are hyper-parameters used to control the impact of the two tasks over the anchor alignment metric $t$. Utilizing the proposed metric $t$, we choose the top $K$ anchor points for supervision at each training step. The part-body association loss is then articulated as:

$$\mathcal{L}_{assoc} = \frac{1}{K} \frac{1}{P} \sum_{\substack{j \in J \\ i \in \{1, \cdot, L\}}} \frac{1}{2} \left( \left\| \lfloor \frac{c_x^b[j]}{s_i} \rfloor - (x_i[j] + \lambda m_i[j]) \right\|_1 + \left\| \lfloor \frac{c_y^b[j]}{s_i} \rfloor - (y_i[j] + \lambda n_i[j]) \right\|_1 \right), \quad (3)$$

with $J$ representing the index list of the top $K$ aligned anchor points for each part and $P$ indicating the number of parts in the image. Thus, the overall loss is expressed as:

$$\mathcal{L} = \lambda_{iou} \mathcal{L}_{iou} + \lambda_{dfl} \mathcal{L}_{dfl} + \lambda_{cls} \mathcal{L}_{cls} + \lambda_{assoc} \mathcal{L}_{assoc}. \quad (4)$$

with $\lambda_{iou}$, $\lambda_{dfl}$, $\lambda_{cls}$, and $\lambda_{assoc}$ are the objective-weighting hyper-parameters.

## 3.4 DECODING PART-TO-BODY ASSOCIATIONS

Our model predicts the center offset between a part and its corresponding body, serving as a crucial link in the part-to-body relationship. During inference, we initiate the process by filtering out overlapping predictions through non-maximum suppression (NMS). This yields refined results for parts and bodies as follows:

$$\hat{O}^b = \text{NMS}(O^b, \tau_{conf}^b, \tau_{iou}^b), \quad \hat{O}^p = \text{NMS}(O^p, \tau_{conf}^p, \tau_{iou}^p), \tag{5}$$

where $\tau_{conf}^b$, $\tau_{conf}^p$, $\tau_{iou}^b$, and $\tau_{iou}^p$ represent the confidence and IoU overlap thresholds for both the body and part in the NMS procedure.

Subsequently, for each part, we compute its anticipated body center using the relationship defined in Equation (2) as:

$$\hat{c}_x^b = s_i(x_i + \lambda m_i), \quad \hat{c}_y^b = s_i(y_i + \lambda n_i). \tag{6}$$

Finally, for each individual part, we determine the Euclidean ($\ell_2$) distance between the estimated body center and the centers of the bodies that are unassigned and also enclose the part. The body with the smallest distance is chosen as the corresponding body for the given part.

The decoding mechanism in our part-to-body association is notably more straightforward than the body-to-part association employed in BPJDet (Zhou et al., 2023b). In BPJDet, complexities arise since each body can be associated with various parts due to occlusions.

## 4 EXPERIMENTS

### 4.1 EVALUATION PROTOCOLS

**Datasets.** We evaluate our method on two datasets: BodyHands (Narasimhaswamy et al., 2022) and COCOHumanParts (Yang et al., 2020). Additional experiments on CrowdHuman (Shao et al., 2018) are provided in Appendix A.1. BodyHands is a large-scale dataset containing unconstrained images with annotations for hand and body locations and correspondences. This dataset consists of 18,861 training images and 1,629 test images. COCOHumanParts is an instance-level human-part dataset with rich-annotated and various scenarios based on COCO 2017 (Lin et al., 2014). This dataset consists of 66,808 training images, 64,115 test images, and 2,693 validation images.

**Evaluation Metric.** We report the standard VOC average precision (AP) metric with IoU $= 0.5$ to evaluate the detection of bodies and parts. We also present the log-average miss rate on false positive per image (FPPI) in the range of $[10^{-2}, 10^0]$ shortened as MR$^{-2}$ (Dollar et al., 2011) of the body and its parts. To qualify the part-body association, we report the log-average miss matching rate (mMR$^{-2}$) on FPPI of part-body pairs in $[10^{-2}, 10^0]$, which was originally proposed by BFJDet (Wan et al., 2021) for exhibiting the proportion of body-face pairs that are mismatched. For BodyHands, we provide the conditional accuracy and joint AP defined by BodyHands (Narasimhaswamy et al., 2022). Lastly, for COCOHumanParts, we follow the evaluation protocols defined in Hier R-CNN (Yang et al., 2020) and report the detection performance with a series of APs (AP$_{.5:.95}$, AP$_{.5}$, A$_M$, A$_L$) as in COCO metrics where subordinate APs represent the part-body association state.

**Implementation Specifics.** Our experimentation framework is constructed using PyTorch (Paszke et al., 2019), and the models are trained across 4 Tesla V100 GPUs, leveraging automatic mixed precision (AMP) over a span of 100 epochs. Consistent with YOLOv7 (Wang et al., 2023a), we utilize the same learning rate scheduler, SGD optimizer (Robbins & Monro, 1951), and prescribed data augmentations. We followe the same protocol established in BPJDet (Zhou et al., 2023b) that uses pretrained YOLO weights. The input image resolution is $1024 \times 1024$, and the batch size is 24. We employ two backbone architectures, YOLOv7 and YOLOv5l6, to demonstrate the versatility and adaptability of our approach.

In our experiments, we set the loss weights $\lambda_{iou} = 7.5$, $\lambda_{dfl} = 1.5$, $\lambda_{cls} = 0.5$, $\lambda_{assoc} = 0.2$, the scaling factor $\lambda = 2.0$, and the anchor alignment parameters $K = 13$, $\alpha = 1.0$, $\beta = 6.0$. During inference, we set the NMS parameters as: the body thresholds $\tau_{conf}^b = 0.05$, $\tau_{iou}^b = 0.6$; while the part thresholds $\tau_{conf}^p = 0.05$, $\tau_{iou}^p = 0.6$ for BodyHands, and $\tau_{conf}^p = 0.005$, $\tau_{iou}^p = 0.75$ for COCOHumanParts.

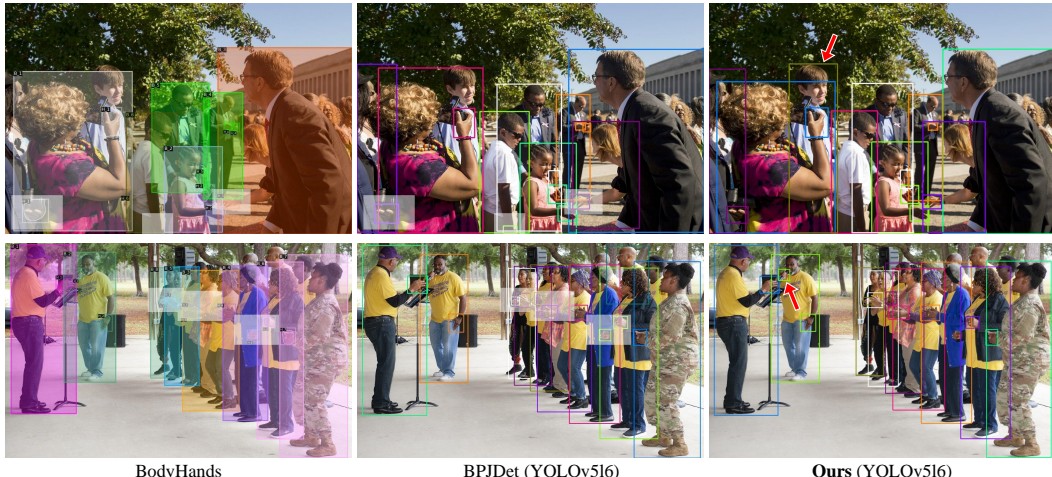

| | BodyHands | BPJDet (YOLOv5l6) | **Ours** (YOLOv5l6) |

Figure 3: Qualitative results on BodyHands. Red arrows highlight our correct predictions.

| Methods | Param (M) | Size | Hand AP↑ | Cond. Accuracy↑ | Joint AP↑ |
|---|---|---|---|---|---|
| OpenPose (2017) | 199.0 | 1536 | 39.7 | 74.03 | 27.81 |
| Keypoint Com. (2021) | 27.3 | 1536 | 33.6 | 71.48 | 20.71 |
| MaskRCNN+FD (2017) | 266.0 | 1536 | 84.8 | 41.38 | 23.16 |
| MaskRCNN+FS (2017) | 266.0 | 1536 | 84.8 | 39.12 | 23.30 |
| MaskRCNN+LD (2017) | 266.0 | 1536 | 84.8 | 72.83 | 50.42 |
| MaskRCNN+IoU (2017) | 266.0 | 1536 | 84.8 | 74.52 | 51.74 |
| BodyHands (2022) | 700.3 | 1536 | 84.8 | 83.44 | 63.48 |
| BodyHands* (2022) | 700.3 | 1536 | 84.8 | 84.12 | 63.87 |
| BPJDet (YOLOv5s6) (2023a) | 15.3 | 1536 | 84.0 | 85.68 | 77.86 |
| BPJDet (YOLOv5m6) (2023a) | 41.2 | 1536 | 85.3 | 86.80 | 78.13 |
| BPJDet (YOLOv5l6) (2023a) | 86.1 | 1536 | 85.9 | 86.91 | 84.39 |
| **Ours** (YOLOv7) | 36.9 | 1024 | **89.1** | 92.62 | **85.98** |
| **Ours** (YOLOv5l6) | 86.1 | 1024 | 88.1 | **92.71** | 85.73 |

Table 1: Quantitative evaluation on BodyHands.

## 4.2 COMPARISON WITH STATE-OF-THE-ART

**BodyHands.** We conduct the hand-body association task on the BodyHands dataset (Narasimhaswamy et al., 2022). Notably, this dataset contains only one type of part annotation since it does not distinguish between left and right hands. Table 1 compares our *PBADet* with leading methods. Our approach surpasses competitors across all metrics, including hand AP, conditional accuracy, and joint AP, by large margins. In particular, our YOLOv7 model, despite having a smaller model size than BPJDet (YOLOv5m6) (Zhou et al., 2023a), delivers superior performance. Furthermore, our YOLOv5l6 model, which utilizes the same backbone as BPJDet (YOLOv5l6), also demonstrates enhanced results. Figure 3 shows that our method demonstrates the best body and part detection and part-body association detection.

**COCOHumanParts.** We evaluate our approach for the multi-part-to-body association task using the COCOHumanParts dataset (Yang et al., 2020), which includes six distinct body parts: head, face, left hand, right hand, left foot, and right foot. Tables 2 and 3 provide a comparative analysis with leading methods. As depicted in Table 2, our method achieves APs that are on par with BJPDet (Zhou et al., 2023b). Notably, we lead in metrics such as $AP_M$, $AP_L$, and APs specific to person and feet detections. Furthermore, Table 3 underscores our method in achieving the lowest average $mMR^{-2}$ across categories like head, face, both left and right hands, and right foot, emphasizing

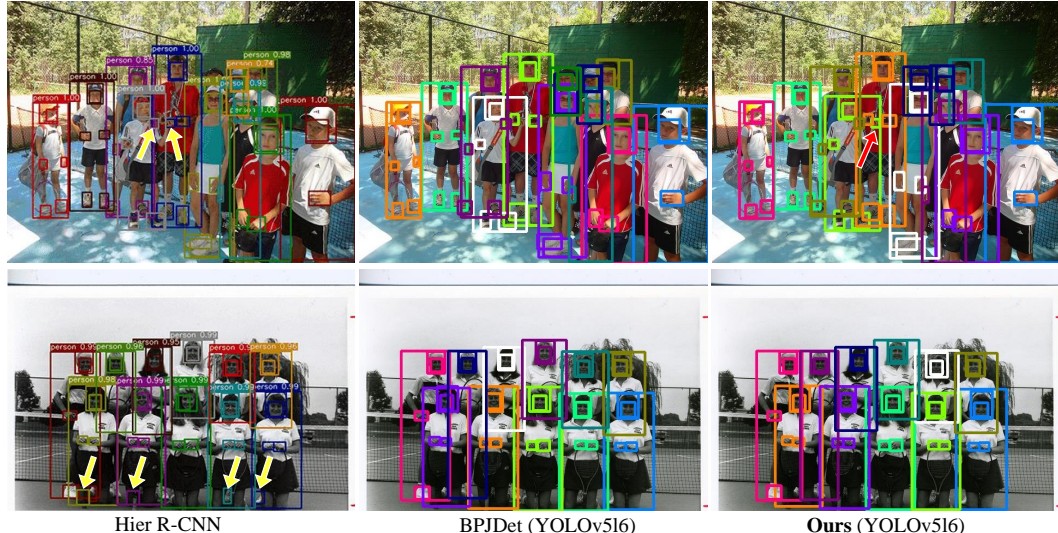

| Hier R-CNN | BPJDet (YOLOv5l6) | **Ours** (YOLOv5l6) |

Figure 4: Qualitative results on COCOHumanParts. Yellow and red arrows highlight other methods' failure cases and our correct predictions, respectively.

| Methods | Joint Detect? | All categories APs↑ (original / subordinate) | | | | Per categories APs↑ | | | | | | |
|---|---|---|---|---|---|---|---|---|---|---|---|---|
| | | $AP_{.5:.95}$ | $AP_{.5}$ | $AP_M$ | $AP_L$ | person | head | face | r-hand | l-hand | r-foot | l-foot |
| Faster-C4-R50 (2015) | | 32.0 / — | 55.5 / — | 54.9 / — | 52.4 / — | 50.5 | 47.5 | 35.5 | 27.2 | 24.9 | 19.2 | 19.3 |
| Faster-FPN-R50 (2017) | ✓ | 34.8 / 19.1 | 60 / 38.5 | 55.4 / 22.4 | 52.2 / 33.6 | 51.4 | 48.7 | 36.7 | 31.7 | 29.7 | 22.4 | 22.9 |
| RetinaNet-R50 (2017b) | | 32.2 / — | 54.7 / — | 54.5 / — | 53.8 / — | 49.7 | 47.1 | 33.7 | 28.7 | 26.7 | 19.7 | 20.2 |
| FCOS-R50 (2019) | | 34.1 / — | 58.6 / — | 55.1 / — | 55.1 / — | 51.1 | 45.7 | 40.0 | 29.8 | 28.1 | 22.2 | 21.9 |
| Faster-FPN-X101 (2017) | | 36.7 / — | 62.8 / — | 57.4 / — | 55.3 / — | 53.6 | 49.7 | 37.3 | 33.8 | 32.2 | 25 | 25.1 |
| Hier-R50 (2020) | ✓ | 36.8 / 33.3 | 65.7 / 67.1 | 53.9 / 29.9 | 47.5 / 47.1 | 53.2 | 50.9 | 41.5 | 31.3 | 29.3 | 25.5 | 26.1 |
| Hier-R101 (2020) | | 37.2 / — | 65.9 / — | 55.1 / — | 50.3 / — | 54.0 | 50.4 | 41.6 | 31.6 | 30.1 | 26.0 | 26.6 |
| Hier-X101 (2020) | ✓ | 38.8 / 36.6 | 68.1 / 69.7 | 56.6 / 32.6 | 52.3 / 51.1 | 55.4 | 52.3 | 43.2 | 33.5 | 32.0 | 27.4 | 27.9 |
| Hier-R50†‡ (2020) | ✓ | 40.6 / 37.3 | 70.1 / 72.5 | 57.5 / 35.4 | 51.5 / 48.9 | — | — | — | — | — | — | — |
| Hier-X101†‡ (2020) | ✓ | 42.0 / 38.8 | 71.6 / 72.3 | 59.0 / 37.4 | 53.3 / 50.3 | — | — | — | — | — | — | — |
| BPJDet (YOLOv5s6) | ✓ | 38.9 / 38.4 | 65.5 / 64.4 | 59.1 / 57.7 | 49.7 / 47.5 | 56.3 | 53.5 | 41.9 | 34.7 | 33.7 | 25.5 | 26.5 |
| BPJDet (YOLOv5m6) | ✓ | 42.0 / 41.7 | 68.9 / 68.1 | 62.3 / 61.6 | 54.6 / 52.9 | 59.8 | 55.7 | 44.7 | 38.7 | 37.6 | 28.6 | 29.2 |
| BPJDet (YOLOv5l6) | ✓ | 43.6 / 43.3 | 70.6 / 69.8 | 63.8 / 63.2 | 61.8 / 58.9 | 61.3 | 56.6 | 46.3 | 40.4 | 39.6 | 30.2 | 30.9 |
| **Ours** (YOLOv7) | ✓ | 42.7 / 41.5 | 69.5 / 67.9 | 65.8 / 64.4 | 66.7 / 63.3 | 62.1 | 54.5 | 44.7 | 38.7 | 37.5 | 30.4 | 30.9 |
| **Ours** (YOLOv5l6) | ✓ | 43.0 / 41.8 | 70.0 / 68.4 | 66.2 / 65.0 | 67.2 / 64.4 | 62.4 | 54.6 | 44.6 | 39.0 | 37.7 | 30.9 | 31.5 |

Table 2: Quantitative evaluation on COCOHumanParts with all and per categories APs. All categories APs include the original detection APs and subordinate APs. The marker †‡ in Hier R-CNN indicates using deformable convolutional layers and multi-scale training (Yang et al., 2020).

our method's superior part-body association capabilities. Figure 4 demonstrates that our method performs as well as (or better than) BPJDet and much better than Hier R-CNN.

**Comparison to Multi-person Pose Estimation.** We present a qualitative analysis contrasting our approach with ED-Pose (Yang et al., 2023), a state-of-the-art multi-person pose estimation method, as illustrated in Figure 5 (see quantitative restuls in Appendix A.2) Despite utilizing the powerful Swin-L backbone (Liu et al., 2021), ED-Pose occasionally provides imprecise or speculative pre-

| Methods | Params (M) | Size | head | face | l-hand | r-hand | r-foot | l-foot |
|---|---|---|---|---|---|---|---|---|
| BPJDet (YOLOv5s6) | 15.3 | 1536 | 33.4 | 36.7 | 50.2 | 51.4 | 58.3 | 58.4 |
| BPJDet (YOLOv5m6) | 41.2 | 1536 | 29.7 | 32.7 | 42.5 | 44.2 | 53.4 | 52.3 |
| BPJDet (YOLOv5l6) | 86.1 | 1536 | 30.8 | 31.8 | 41.6 | 42.2 | 49.1 | 50.4 |
| **Ours** (YOLOv7) | 36.9 | 1024 | 29.2 | 31.3 | 41.1 | 42.7 | 52.6 | 52.5 |
| **Ours** (YOLOv5l6) | 86.1 | 1024 | 27.9 | 30.5 | 36.2 | 40.8 | 50.4 | 47.9 |

Table 3: Quantitative evaluation on COCOHumanParts with the average $mMR^{-2}$.

dictions, especially for limbs such as legs and arms. Such unreliable results cloud the discernment of the relationship between a body and its corresponding parts. Our part-body association detention method can effectively address and diminish this issue.

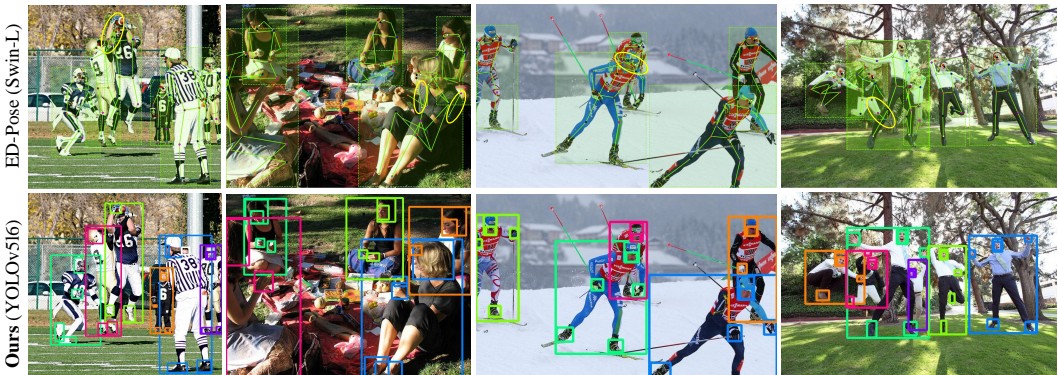

Figure 5: Qualitative comparison with ED-Pose (Yang et al., 2023) on images from COCO. Yellow circles highlight erroneous predictions.

### 4.3 ABLATION STUDY

| Methods | Hand AP↑ | Cond. Accuracy↑ | Joint AP↑ |
|---|---|---|---|
| w/o $\mathcal{L}_{assoc}$ (baseline) | **89.1** | 80.78 | 78.07 |
| w/o Multi-scale | 88.8 | 91.64 | 85.46 |
| w/o Task-align | 89.0 | 92.08 | 85.78 |
| Full | **89.1** | **92.62** | **85.98** |

Table 4: Ablation experiments on BodyHands with the YOLOv7 backbone architecture.

To better understand the contribution of various components in our approach, we undertake ablation experiments on the BodyHands dataset with the YOLOv7 backbone architecture, summarized in Table 4 and more in Table 8 in Appendix A.4. Initially, we examine a configuration without the association loss, $\mathcal{L}_{assoc}$, which serves as our baseline for detecting human bodies and parts. For part-body association in this scenario, we utilize Euclidean distances between a part center and the unassigned body centers. The body that encloses the part and has the minimal distance is then tagged as the corresponding body for the given part. Compared to the baseline model, the introduction of our association prediction head maintains the accuracy in hand detection, and considerably boosts the efficacy of part-body association.

In our subsequent variant, the center offset is defined in relation to the multi-scale feature anchor points. When this center offset is interpreted as a normalized absolute distance (*i.e.*, *w/o Multi-scale*), we observe a decline in metrics, including hand prediction accuracy, conditional accuracy, and joint AP. Lastly, our method adopts task alignment learning for supervising part-body associations. Disregarding task alignment (*i.e.*, *w/o Task-align*), where all positive samples are used for training, leads to decreased performance in all metrics, in contrast with our full model.

## 5 CONCLUSION

The task of human part-body detection and association is critical for a wide range of computer vision applications. In this work, we introduced an innovative one-stage, anchor-free methodology for this purpose. By extending the anchor-free object representation across multi-scale feature maps, we incorporate a singular part-to-body center offset, effectively bridging parts and their corresponding bodies. Our approach is purposely designed to handle multiple parts-to-body associations without sacrificing detection accuracy or robustness. Experimental results validate that our proposed method sets new standards in the domain, surpassing existing state-of-the-art solutions.

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

| Methods | Stage | MR$^{-2}$↓ body | MR$^{-2}$↓ face | AP↑ body | AP↑ face | mMR$^{-2}$↓ body-face |
|---|---|---|---|---|---|---|
| RetinaNet+POS (2017b) | One | 52.3 | 60.1 | 79.6 | 58.0 | 73.7 |
| RetinaNet+BFJ (2017b) | One | 52.7 | 59.7 | 80.0 | 58.7 | 63.7 |
| FPN+POS (2017a) | Two | 43.5 | 54.3 | 87.8 | 70.3 | 66.0 |
| FPN+BFJ (2017a) | Two | 43.4 | 53.2 | 88.8 | 70.0 | 52.5 |
| CrowdDet+POS (2020) | Two | 41.9 | 54.1 | 90.7 | 69.6 | 64.5 |
| CrowdDet+BFJ (2020) | Two | 41.9 | 53.1 | 90.3 | 70.5 | 52.3 |
| BPJDet (YOLOv5s6) (2023a) | One | 41.3 | 45.9 | 89.5 | 80.8 | 51.4 |
| BPJDet (YOLOv5m6) (2023a) | One | 39.7 | 45.0 | 90.7 | 82.2 | 50.6 |
| BPJDet (YOLOv5l6) (2023a) | One | 40.7 | 46.3 | 89.5 | 81.6 | 50.1 |
| **Ours** (YOLOv7) | One | 36.5 | 44.7 | 91.6 | 81.6 | 50.8 |
| **Ours** (YOLOv5l6) | One | 37.5 | 45.3 | 91.4 | 81.2 | 50.9 |

Table 5: Quantitative evaluation on CrowdHuman for the face-body association task.

| Methods | Joint Detect? | Backbone | body MR$^{-2}$↓ | body AP↑ | mMR$^{-2}$↓ |
|---|---|---|---|---|---|
| Adaptive-NMS (2019) | ✗ | CNN | 49.7 | 84.7 | — |
| PBM (2020) | ✗ | CNN | 43.3 | 89.3 | — |
| CrowdDet (2020) | ✗ | CNN | 41.4 | 90.7 | — |
| AEVB (2021) | ✗ | CNN | 40.7 | — | — |
| AutoPedestrian (2021) | ✗ | CNN | 40.6 | — | — |
| Beta RCNN(KL$_{th}$ = 7) (2020) | ✗ | CNN | 40.3 | 88.2 | — |
| PedHunter (2020a) | ✗ | CNN | 39.5 | — | — |
| Sparse-RCNN (2021) | ✗ | DETR | 44.8 | 91.3 | — |
| Deformable-DETR (2020) | ✗ | DETR | 43.7 | 91.5 | — |
| PED-DETR (2020) | ✗ | DETR | 43.7 | 91.6 | — |
| Iter-E2EDET (2022) | ✗ | DETR | 41.6 | 92.5 | — |
| DA-RCNN (2019) | ✓ | CNN | 52.3 | — | — |
| DA-RCNN + J-NMS (2019) | ✓ | CNN | 51.8 | — | — |
| JointDet (2020b) | ✓ | CNN | 46.5 | — | — |
| BPJDet (YOLOv5s6) (2023b) | ✓ | CNN | 41.3 | 89.5 | 51.4 |
| BPJDet (YOLOv5m6) (2023b) | ✓ | CNN | 39.7 | 90.7 | 50.6 |
| BPJDet (YOLOv5l6) (2023b) | ✓ | CNN | 40.7 | 89.5 | 50.1 |
| **Ours** (YOLOv7) | ✓ | CNN | 36.5 | 91.6 | 50.8 |
| **Ours** (YOLOv5l6) | ✓ | CNN | 37.5 | 91.4 | 50.9 |

Table 6: The performance comparison of our method for the joint *face-body* detection task with other crowded person detection methods in the val-set of CrowdHuman.

# A  APPENDIX

## A.1  EXPERIMENTS ON CROWDHUMAN

We additionally conduct experiments on the CrowdHuman dataset (Shao et al., 2018). CrowdHuman is a dataset tailored for crowded scenarios, providing individual annotations for each pedestrian. This dataset includes 15,000 training images and 4,375 validation images. The initial annotations include boxes for visible bodies and heads. Labels of faces are added by BFJDet (Wan et al., 2021) for the body-face association.

In our experiments, the input image resolution is $1536 \times 1536$, and the batch size is 12. The other settings are the same as in the main paper. During inference, we set the NMS parameters as: the body thresholds $\tau_{conf}^b = 0.05$, $\tau_{iou}^b = 0.6$; while the part thresholds $\tau_{conf}^p = 0.1$, $\tau_{iou}^p = 0.3$ as in BPJDet (Zhou et al., 2023b).

We benchmark our approach for the face-body association task utilizing the CrowdHuman dataset (Shao et al., 2018). The results, as illustrated in Table 5, draw a comparison between our proposed *PBADet* and existing methods. Performance-wise, *PBADet* is comparable with BPJDet, with our *PBADet* (YOLOv7) achieving the best metrics in $MR^{-2}$ for both body and face, as well as AP for the body. It's essential to highlight that the face-body association task is singular in nature, involving just one part. Consequently, the advantages of our method over BPJDet are not as pronounced as seen in scenarios with multiple parts-to-body associations, such as on the BodyHands and COCO-HumanParts datasets.

Besides, we also try to compare our joint detector *PBADet* with other methods specifically designed for crowded person detection, either CNN-based or DETR-based (Carion et al., 2020). As shown in Table 6, apart from obtaining the best $MR^{-2}$ result for body detection, our *PBADet* achieves a comparable AP performance with those cumbersome DETR-based detectors such as PED-DETR (Lin et al., 2020) and Iter-E2EDET (Zheng et al., 2022). Comparing with the counterpart BPJDet, our superiority is still evident. This indicates that our proposed *PBADet* for addressing the face-body association task can keep a better balance between detection and association.

## A.2 QUANTITATIVE COMPARISON TO MULTI-PERSON POSE ESTIMATION

| Parts | ED-Pose (2023) | | | Ours | | |
|---|---|---|---|---|---|---|
| | Precision↑ | Recall↑ | F1↑ | Precision↑ | Recall↑ | F1↑ |
| *lefthand* | 18.36 | 30.34 | 22.87 | **37.63** | **60.82** | **46.50** |
| *righthand* | 19.57 | 31.59 | 24.17 | **38.58** | **61.26** | **47.34** |
| *leftfoot* | 29.32 | 65.28 | 40.46 | **41.56** | **66.24** | **51.08** |
| *rightfoot* | 29.13 | 65.58 | 40.34 | **40.86** | **65.55** | **50.34** |
| *all parts* | 24.09 | 45.51 | 31.51 | **39.41** | **63.09** | **48.52** |

Table 7: Comparison with ED-Pose (Yang et al., 2023), a state-of-the-art multi-person pose estimation method.

We compare our method with ED-Pose (Yang et al., 2023), a current leading method in multi-person pose estimation. A qualitative comparison with ED-Pose on images from COCO (Lin et al., 2014) is presented in Figure 5.

For a fair and meaningful quantitative comparison, we conducted evaluations on the COCO validation set (Lin et al., 2014), as ED-Pose was also trained on the COCO dataset. This validation set contains 2,693 images, each featuring at least one person. To ensure consistency, we utilized the ground truth body-part bounding boxes from COCO-HumanParts, which are derived from the original COCO annotations. We excluded facial and head parts due to their inconsistent number of keypoints and focused on four challenging and smaller parts: left hand, right hand, left foot, and right foot, which can be determined by one keypoint. For evaluation, we used the ED-Pose model (Swin-L) trained on the COCO training set and our PBADet model (YOLOv5l6) trained on the COCO-HumanParts training set.

In assessing performance, we considered a predicted keypoint by ED-Pose as a true positive if it fell within a ground truth part box. In favor of ED-Pose, we do not use a part detection model but directly use the ground truth part boxes. For PBADet, a predicted part box with successful part-body association was deemed a true positive when it achieved an IoU greater than 0.5 with the ground truth box. The results, as shown in the table below, indicate that PBADet demonstrates superior robustness in part-body association compared to ED-Pose, especially in the precision, recall, and F1 score metrics for the selected body parts. This comparative analysis underscores the efficacy of PBADet in handling part-body associations, affirming its robustness and precision over existing methods in this domain.

## A.3 EXPERIMENTS ON ANIMAL DATASETS

We expand our research to include experiments on AnimalPose (Cao et al., 2019) and AP-10K (Yu et al., 2021) to demonstrate our model's scalability and performance in domains beyond human part-to-body associations.

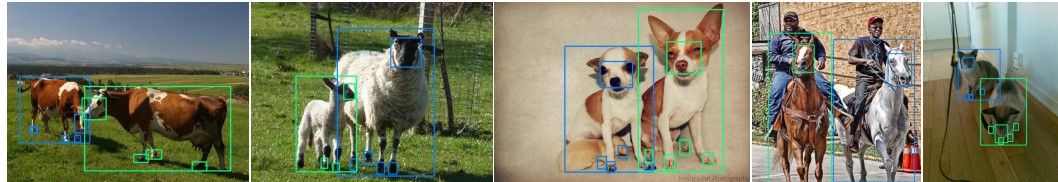

Figure 6: Qualitative results on AP-10K (Yu et al., 2021).

Our study includes five common quadruped animals: dogs, cats, sheep, horses, and cows. These animals are chosen due to the similarity in their anatomical structure, particularly having five identifiable parts: head and four feet. This similarity allows us to treat these diverse animal types as a single class for analysis purposes. The body bounding boxes for these animals are derived from the provided ground truth labels in the datasets, while the part bounding boxes are generated based on keypoint annotations. In our experiments, we use 4,608 images with 6,117 instances of animal bodies from AnimalPose (Cao et al., 2019) for training and use 2,000 images containing 7,962 body instances from AP-10K (Yu et al., 2021) for evaluation. We train our PBADet (YOLOv5l6) model with the same experimental setup as described in Section 4.1 of the main paper.

Qualitative results, as illustrated in Figure 6, showcase successful part-body association in these animals. These outcomes affirm that extending our method to quadruped animals not only is feasible but also yields convincing and meaningful results, demonstrating the potential of our approach for broader applications in part-body association tasks across different domains.

## A.4 MORE ABLATION STUDY

| Methods | Param (M) | Size | Hand AP↑ | Cond. Accuracy↑ | Joint AP↑ |
|---|---|---|---|---|---|
| BPJDet (anchor-based body-to-part) | 41.2 | 1536 | 85.3 | 86.80 | 78.13 |
| **Ours** (anchor-based part-to-body) | 36.9 | 1024 | 88.4 | 92.31 | 85.28 |
| **Ours** (anchor-free part-to-body) | 36.9 | 1024 | **89.1** | **92.62** | **85.98** |

Table 8: Comparison of anchor-based vs. anchor-free backbones and body-to-part vs. part-to-body definitions.

We conduct an ablation study to compare our model with the traditional anchor-based YOLOv7 model, using the same association definition of part-to-body center offset as in our approach. This comparison has been instrumental in highlighting the advantages of the anchor-free design, especially in the context of part-body association tasks. Our findings show that the anchor-free version of YOLOv7 outperforms its anchor-based counterpart, validating the effectiveness of our method. Furthermore, we compared the traditional anchor-based YOLOv7 model with BPJDet, which also adopts an anchor-based approach but utilizes a body-to-part association definition. This comparison demonstrates that our part-to-body association definition significantly enhances performance, offering clear evidence of the superiority of our approach over previous methods.

## A.5 DETAIL RESULTS OF COCOHUMANPARTS

| | All categories APs↑ (original / subordinate) | | | | | |
|---|---|---|---|---|---|---|
| | $AP_{.5:.95}$ | $AP_{.5}$ | $AP_{.75}$ | $AP_S$ | $AP_M$ | $AP_L$ |
| **Ours** (YOLOv7) | 42.7 / 41.5 | 69.5 / 67.9 | 43.5 / 42.2 | 31.4 / 30.3 | 65.8 / 64.4 | 66.7 / 63.3 |
| **Ours** (YOLOv5l6) | 43.0 / 41.8 | 70.0 / 68.4 | 43.7 / 42.4 | 31.5 / 30.3 | 66.2 / 65.0 | 67.2 / 64.4 |

Table 9: Evaluation results of the proposed method on COCOHumanParts with all categories APs.

In the original Hier R-CNN paper (Yang et al., 2020), small objects are further categorized into small and tiny objects, and AP for small objects is not explicitly provided. Similarly, another comparison

method, BPJDet (Zhou et al., 2023b), also does not offer AP details for small objects. We further include the results of $AP_{.75}$ and $AP_S$ in Table 9 for transparency and comparison to future work. We emphasize that Table 2 demonstrates our method's competitive object detection accuracy relative to state-of-the-art approaches. More importantly, the critical metric for our task is association accuracy, where, as shown in Table 3, our method achieves superior performance.

