# OpenReview forum: "PBADet: A One-Stage Anchor-Free Approach for Part-Body Association"
_ICLR.cc/2024/Conference — ICLR 2024 poster_

### Official Review · Reviewer_iAuQ · 2023-10-17

**Soundness:** 3 good
**Presentation:** 3 good
**Contribution:** 2 fair
**Rating:** 5
**Confidence:** 4

**Summary:**

This paper propose an association method for part-body detection. Their framework is based on YOLO, which is anchor-free.  In experiments, they show PBADet's good performance.

**Strengths:**

1. This paper is easy to read.
2. Association for body and parts is  worth studying.
3. The method is evaluated in several datasets.

**Weaknesses:**

1.The main problem of this paper is that part-to-body offsets have been proposed in previous works [1]. Furthermore，in [1], they also try local center offsets to improve the performance. Hence, considering the not new idea and no other designs, the technical learning of this paper is limitted.
[1] Jin L, Wang XJ. Grouping by Center: Predicting Centripetal Offsets for the Bottom-up Human Pose Estimation. TMM, 2022.
2. "Single-stage" is controverial. This method is bottom-up but still need two-stage processing, i.e., detection and grouping. And single-stage method has been studied in several reaseraches [2]. There are  obvious differences between single-stage methods and this paper.
[2] Single-Stage Multi-Person Pose Machines, ICCV, 2019.
3. To verify the association efficiency, author should compare your association method with other grouping methods, such as associative embeddings and methods in [1], under the same part detection accuracy.
[3] Associative Embedding: End-to-End Learning for Joint Detection and Grouping.
4. The focus of this paper is association, but the evaluation metric is AP. More metrics on association quality should be discussed.

**Questions:**

Refer to the weakness.

---

> ### Author Response · Authors · 2023-11-14
>
> Dear Reviewer iAuQ,
>
> Thank you for your detailed review and for raising critical points about our paper. Your feedback is essential for refining our work, and we have addressed each of your concerns below.
>
> 1. Originality of Part-to-Body Offsets:
>
> A:  Thank you for highlighting the paper [1] that discusses a concept similar to our part-to-body center offset. However, it is crucial to note the distinct focus of our work on the part-body association task, as opposed to the multi-person pose estimation task emphasized in [1]. This distinction is elaborated in Section 2 "Related Work" of our paper.
>
> The part-body association task, exemplified by models like BodyHands, Hier R-CNN, and BPJDet, involves detecting the bounding boxes of both parts and bodies and establishing their association relationships. In contrast, multi-person pose estimation primarily focuses on detecting individual keypoints of each person.
>
> To illustrate the practical application of our approach, consider hand gesture recognition in a scan room setting. Here, it is essential for the system to respond exclusively to the technician's gestures while ignoring the patient’s hands to prevent unintended interactions. This scenario requires not just the detection of hands (parts) but also the identification of their associated body, necessitating bounding boxes for both hand and body for effective gesture recognition and person classification.
>
> As discussed in [4], while multi-person pose estimation can be adapted for the part-body association task, it requires additional models for detecting part and body bounding boxes. Using pose estimation for association, i.e., determining whether a keypoint lies within a part's bounding box, introduces a two-stage heuristic process that can add complexity and potentially reduce efficiency. Our method addresses this gap by providing a more streamlined, single-stage solution for the part-body association, enhancing both the efficiency and applicability of the approach in practical scenarios.
>
> [4] Who are raising their hands? Hand-raiser seeking based on object detection and pose estimation
>
> 2. Definition of 'Single-Stage' Method:
>
> A: In addressing your point about the classification of our method as 'single-stage', particularly in the context of multi-person pose estimation, we acknowledge that there could be a perspective from which it might be seen as a two-stage method. However, our definition of 'stages' is centered around the process of association prediction. We follow the definition of stages in other hand-body association works such as BPJDet, which is considered to be single-stage.
>
> In earlier works like [4] and BodyHands [5], the task of association prediction typically involves two distinct stages. The first stage is the detection of part and body bounding boxes, and the second stage encompasses either employing pose estimation with a heuristic strategy or using a learned association network to link the part and body. This two-stage process, though effective, introduces additional complexity and potential inefficiencies.
>
> Contrasting this, both our method and BPJDet streamline the association task. These models are designed to simultaneously output the part and body bounding boxes along with their association vectors. This integration significantly simplifies the process, effectively making it a single-stage approach in terms of association prediction. By directly outputting both detection and association information, our model avoids the complexities inherent in traditional two-stage methods, thereby enhancing efficiency and reducing computational overhead.
>
> [5] Whose hands are these? Hand detection and hand-body association in the wild
>
> **Continue in the next comment**

---

> > ### Author Response · Authors · 2023-11-14
> >
> > 3. Comparison with Multi-Person Pose Estimation Methods:
> >
> > A: We thank you for suggesting relevant works for comparison. In our revised paper, we will include discussions on these works, particularly [1] and [3], in the Related Work section. Considering that [1] is not open-sourced and [3] was proposed in 2017 with performance no longer considered state-of-the-art, we have chosen to compare our method with ED-Pose [6], a current leading method in multi-person pose estimation. We have presented a qualitative comparison with ED-Pose on images from COCO in Figure 5.
> >
> > For a fair and meaningful quantitative comparison, we conducted evaluations on the COCO val-set, as ED-Pose was also trained on the COCO dataset. This validation set contains 2,693 images, each featuring at least one person. To ensure consistency, we utilized the ground truth body-part bounding boxes from COCO-HumanParts, which are derived from the original COCO annotations. We excluded facial and head parts due to their inconsistent number of keypoints and focused on four challenging and smaller parts: left hand, right hand, left foot, and right foot, which can be determined by one keypoint. For evaluation, we used the ED-Pose model (Swin-L) trained on the COCO train-set and our PBADet model (YOLOv5l6) trained on the COCO-HumanParts train-set.
> >
> > In assessing performance, we considered a predicted keypoint by ED-Pose as a true positive if it fell within a ground truth part box. In favor of ED-Pose, we do not use a part detection model but directly use the ground truth part boxes. For PBADet, a predicted part box with successful part-body association was deemed a true positive when it achieved an IoU greater than 0.5 with the ground truth box. The results, as shown in the table below, indicate that PBADet demonstrates superior robustness in part-body association compared to ED-Pose, especially in the precision, recall, and F1 score metrics for the selected body parts. This comparative analysis underscores the efficacy of PBADet in handling part-body associations, affirming its robustness and precision over existing methods in this domain.
> >
> > | Methods                  | Parts     | Precision | Recall | F1   |
> > |--------------------------|-----------|-----------|--------|------|
> > |                          | lefthand  | 18.36     | 30.34  | 22.87|
> > |                          | righthand | 19.57     | 31.59  | 24.17|
> > | ED-Pose                  | leftfoot  | 29.32     | 65.28  | 40.46|
> > |                          | rightfoot | 29.13     | 65.58  | 40.34|
> > |                          | all parts | 24.09     | 45.51  | 31.51|
> > |--------------------------|-----------|-----------|--------|------|
> > |                          | lefthand  | 37.63     | 60.82  | 46.50|
> > |                          | righthand | 38.58     | 61.26  | 47.34|
> > | Ours                     | leftfoot  | 41.56     | 66.24  | 51.08|
> > |                          | rightfoot | 40.86     | 65.55  | 50.34|
> > |                          | all parts | 39.41     | 63.09  | 48.52|
> >
> >
> > [6] Explicit box detection unifies end-to-end multi-person pose estimation
> >
> > 4. Evaluation Metrics for Association:
> >
> > A: In Section 4.1 "EVALUATION PROTOCOLS", we outlined various metrics for assessing association accuracy. Alongside AP, we utilize Conditional Accuracy (defined as the percentage of correctly associated bodies among the correctly detected hand instances) and Joint AP, as defined by BodyHands, and log-average miss matching rate (mMR−2) on FPPI of part-body pairs, as defined by BPJDet (please refer to the original papers for detailed definitions). These metrics, presented in Tables 1, 4, 3, 5, and 6, offer a comprehensive assessment of our method's association accuracy.
> >
> > We hope these revisions address your concerns and provide a more comprehensive understanding of our method and its unique contributions. We are dedicated to making the necessary updates to enhance the clarity and impact of our paper.

---

> > ### Comment · Reviewer_iAuQ · 2023-11-17
> > **About the originality**
> >
> > In your answer, you only state the difference between the two tasks, i.e., pose estimation and part-body association. I do not deny the two tasks have differences and also share some commons. However, the main problem is the same idea of your work and the reference [1], which is using centripetal offset as a grouping clues. Furthermore, with the same idea, you have not proposed other modules or designs to give more lights for this problem.
> >
> > Even if your paper is accepted, I hope you can discuss [1] in related work.

---

> > > ### Author Response · Authors · 2023-11-20
> > >
> > > Dear iAuQ,
> > >
> > > Thank you for your continued engagement and insightful comments. In our revised manuscript, we will indeed include a detailed discussion of references [1], [2], and [3] in the Related Work section to provide a comprehensive context, as we stated in our previous answer 3. While we acknowledge the shared use of centripetal offsets, it is crucial to emphasize the distinct functionalities and roles these offsets play in our respective studies.
> > >
> > > In the reference paper [1], centripetal offsets are primarily employed for grouping human joints, aiding in the efficient and accurate estimation of multi-person poses. This technique focuses on solving the complex task of pose estimation in crowded scenes by linking keypoints to a common center - the body center. Here, the offsets serve as a means to aggregate disparate keypoints into identifiable human figures.
> > >
> > > Contrastingly, in our work, the part-to-body center offset has a fundamentally different role and functionality. Our focus is on the precise and specific association between individual body parts and the corresponding bodies, crucial for applications requiring accurate and unambiguous part-body relationships, such as in gesture recognition systems. The part-to-body center offset in our model is not just a grouping mechanism but a critical component for establishing a **one-to-one correspondence** between parts and bodies. This approach contrasts with the one-to-many correspondence of the body-to-part association as represented in BPJDet, effectively ensuring precise and accurate association of each part with its specific corresponding body.
> > >
> > > This difference in application scope and functionality—from grouping for pose estimation in [1] to one-to-one corresponding for precise part-body association in our work—highlights the distinct challenges that our method addresses. Our approach is tailored for scenarios that require a high degree of specificity and accuracy in part-to-body associations, a need not directly addressed by the methodology in [1] where the offsets serve as a means to group disparate keypoints for multi-person poses.
> > >
> > > Given these distinctions in functionality and application, we kindly request a reconsideration of your evaluation. Our work contributes uniquely to the field, addressing specific challenges in part-body association distinct from those in human pose estimation. We hope our forthcoming revisions will further illuminate these unique contributions, aligning our work with the esteemed standards of the conference.
> > >
> > > We appreciate your thoughtful consideration.

---

### Official Review · Reviewer_vvPE · 2023-10-29

**Soundness:** 2 fair
**Presentation:** 1 poor
**Contribution:** 3 good
**Rating:** 6
**Confidence:** 5

**Summary:**

This article presents a method for correlating part to body relationships in a single multi-person RGB image. The method is a single-stage approach that adopts the popular center-based detection architecture. To correlate a subject and multiple detected body parts, a common approach is to estimate center offsets from each body center to the underlying body part. But this may be ineffective since in case of occlusion/close interaction multiple body centers may point to the same body part. Therefore, this paper proposes to simultaneously estimate the offset of the body part position from the detected body part toward the center of the subject. This means that instead of estimating center-to-part offsets, estimating part-to-center offsets for each detected body part will avoid ambiguities in close interaction situations. The proposed method greatly improves the performance of hand association in BodyHands.

**Strengths:**

1. BodyHands performance improvements are impressive with simple associative design changes. Changing from center-to-part to part-to-center offset greatly alleviates the ambiguity of hand associations in training.

2. Interesting insights. The proposed method can achieve significant performance improvements in hand association on BodyHands. This demonstrates the interesting insight that the position of the hand is more ambiguous than the position of the center of the body. Because more occlusion/interaction occurs in the hand area.

**Weaknesses:**

1. Writings. It is important to clearly present ideas and implementation. The current presentation is very vague and difficult to understand. It is beneficial to emphasize conceptual-level differences in the introduction. But before that, the paper should at least introduce the method/idea clearly. Additionally, in the Methods and related work section, consider clearly presenting differences compared to previous methods.

2. In Figure 2, please highlight the parts where the proposed approach makes design changes. In Fig. 1, the direction of arrows are not very obvious. Besides, differences in qualitative results are difficult to distinguish. Please consider emphasizing it further.

**Questions:**

1. Will the proposed method be open-sourced? It seems hard to re-implement the results with limited details.

---

> ### Author Response · Authors · 2023-11-14
>
> Dear reviewer vvPE:
>
> Thank you for your thorough review and constructive feedback. We appreciate your emphasis on clarity in presentation and the importance of detailed documentation for reproducibility. Please find below our responses to your comments and queries.
>
> 1. Improving Clarity in Writing:
>
> A: We acknowledge your concerns regarding the clarity of presentation in our paper, particularly in the Methods and Related Work sections. We understand the importance of clearly articulating the conceptual differences between our method and previous methods. We will undertake a thorough revision of these sections to enhance clarity and ensure that our method and its unique aspects are presented in a more comprehensible and detailed manner.
>
> 2. Revising Figures for Enhanced Clarity:
>
> A: We appreciate your suggestions regarding Figure 2 and Figure 1. In the revised version of the paper, we will highlight the parts in Figure 2 where our approach introduces design changes. We will also address the clarity of the arrows in Figure 1 and enhance the differentiation in qualitative results to ensure that the unique aspects of our approach are more evident.
>
> 3. Availability of Code and Re-Implementation Details:
>
> A: We have outlined the primary details under Section 4.1 "EVALUATION PROTOCOLS" in the main paper. Regarding the availability of our code, we are in the process of finalizing its release, pending approval. We are actively working towards this goal and plan to update the revised paper accordingly. Furthermore, we are committed to providing even more detailed implementation to ensure that other researchers and practitioners can reliably reproduce our results.
>
> We are grateful for your feedback, as it plays a crucial role in refining our work. We are dedicated to addressing these points comprehensively in the revised version of our paper.

---

### Official Review · Reviewer_TS1Z · 2023-10-30

**Soundness:** 2 fair
**Presentation:** 3 good
**Contribution:** 3 good
**Rating:** 6
**Confidence:** 4

**Summary:**

This paper presents PBADet, a one-stage, anchor-free object detection model that efficiently identifies and associates object parts, such as human body parts, to their main body. Utilizing multi-scale feature maps and a unique part-to-body center offset, PBADet offers improved accuracy and robustness over existing models. It addresses inefficiencies through a task-aligned learning strategy and a simplified decoding process for part-to-body associations, making it a streamlined and effective solution in object detection.

**Strengths:**

1. The framework is adaptable for various part-to-body association challenges beyond human body parts.
2. The model adopts an anchor-free paradigm, potentially improving detection performance for objects of varying sizes.
3. Incorporation of a task-alignment learning strategy for bounding box, class predictions, and part-body association.

**Weaknesses:**

1. The paper doesn't explicitly address how the model deals with occluded parts.
2. The dense, per-pixel prediction approach might lead to increased computational demands.
3. The performance might be highly sensitive to the tuning of loss function hyper-parameters.
4. There's a lack of in-depth comparative analysis with traditional anchor-based models.
5. The model's efficiency in handling images with multiple overlapping objects is unclear.

**Questions:**

1. How does PBADet address the challenge of occluded parts in the detection process?
2. How does the model perform in scenarios with multiple overlapping or closely situated objects?
3. Have you tested the model’s scalability and performance across various domains other than human part-to-body associations?
4. Could you provide more comparative performance of PBADet against traditional anchor-based detection models?

---

> ### Author Response · Authors · 2023-11-14
>
> Dear Reviewer TS1Z:
>
> Thank you for your detailed review and insightful queries regarding our paper. Your feedback is invaluable in improving our work. Below, we address each of your points to clarify our methodology and findings.
>
> 1. Handling of Occluded Parts:
>
> A: Our method does not explicitly address occluded parts. For instance, in hand-body associations, if a hand is not visible (occluded), it is not associated with a body. This approach demonstrates an advantage over methods like BPJDet, which outputs a constant number of part center offsets regardless of visibility, making the association decoding more complex. Our part-to-body center offset definition ensures that no offset is attached to undetected (invisible) parts, enhancing the method's relevance and efficiency.
>
> 2. Dense Per-Pixel Prediction and Computational Efficiency:
>
> A: The one-stage anchor-free object detection approach that we employ, with its dense per-pixel prediction, is established for real-time performance on many platforms. It is more efficient than many two-stage proposal-based object detection methods and traditional anchor-based models, which require multiple predefined anchor boxes for each prediction. Please refer to our response to Reviewer AkLB: "3. Model Capacity and Computational Complexity".
>
> 3. Sensitivity to Loss Function Hyper-Parameters:
>
> A: We have calibrated the association loss weight (λ_assoc=0.2) to ensure that all losses are of the same magnitude. Other hyper-parameters are set as defaults in established object detection models. Note that, we maintain these parameters consistently across different datasets to demonstrate the generalizability of our method. For NMS thresholds, we select optimal values for each dataset considering their specific levels of crowdedness, which is in line with protocols followed in previous methods, such as BPJDet, ensuring a fair and effective comparison.
>
> 4. Comparative Analysis with Anchor-Based Models:
>
> A: We acknowledge the importance of an in-depth comparative analysis with traditional anchor-based models. BPJDet, a state-of-the-art anchor-based method, has already been compared in our paper. Additionally, we have introduced a new ablation study in our revised paper to compare our model with the traditional anchor-based YOLOv7 model, using the same association definition of part-to-body center offset as in our approach. This comparison has been instrumental in highlighting the advantages of the anchor-free design, especially in the context of part-body association tasks. Our findings show that the anchor-free version of YOLOv7 outperforms its anchor-based counterpart, validating the effectiveness of our method. Furthermore, we compared the traditional anchor-based YOLOv7 model with BPJDet, which also adopts an anchor-based approach but utilizes a body-to-part association definition. This comparison demonstrates that our part-to-body association definition significantly enhances performance, offering clear evidence of the superiority of our approach over previous methods.
>
> | Methods                  | Param (M) | Size | Hand AP↑ | Cond. Accuracy↑ | Joint AP↑ |
> |--------------------------|-----------|------|----------|-----------------|-----------|
> | BPJDet (anchor-based body-to-part)    | 41.2      | 1536 | 85.3     | 86.80           | 78.13     |
> | Ours (anchor-based part-to-body)     | 36.9      | 1024 | 88.4     | 92.31           | 85.28     |
> | Ours (anchor-free part-to-body)      | 36.9      | 1024 | 89.1     | 92.62           | 85.98     |
>
>
> 5. Applicability Across Various Domains:
>
> A: We have expanded our research to include experiments on the AnimalPosedataset [1] [2], demonstrating our model's scalability and performance in domains beyond human part-to-body associations. The qualitative results of these experiments are included in the revised paper.
>
> [1] Cross-Domain Adaptation for Animal Pose Estimation
>
> [2] AP-10K: A Benchmark for Animal Pose Estimation in the Wild
>
> 6. Performance in Scenarios with Overlapping or Closely Situated Objects:
>
> A: We have conducted experiments on the COCOHumanParts dataset, which includes scenarios where people overlap and are closely situated, as illustrated in Figure 4. Additionally, our Appendix includes experiments on the CrowdHuman dataset, featuring more densely overlapping situations.
>
> We hope these responses address your concerns and provide a clearer understanding of our approach and its capabilities. We are committed to making the necessary revisions to reflect these clarifications and enhance the overall quality of our paper.

---

> > ### Comment · Reviewer_TS1Z · 2023-11-20
> >
> > I have read the author's rebuttal feedback as well as other reviews. The rebuttal has answered many of my questions, and I have increased my original rating score.

---

### Official Review · Reviewer_NDUe · 2023-10-31

**Soundness:** 3 good
**Presentation:** 3 good
**Contribution:** 2 fair
**Rating:** 6
**Confidence:** 4

**Summary:**

The paper addresses the problem of human body part detection and association. The authors present a one-stage, anchor-free method to tackle this problem. Specifically, they introduce a part-to-body center offset to capture the relationship between parts and their corresponding bodies. The authors conduct experiments on BodyHands, COCOHumanParts, and CrowdHuman datasets.

**Strengths:**

- The paper is well-written and easy to follow.
- The proposed single-part-body center offset is efficient and can accommodate a large number of body parts without increasing the number of offsets as the number of parts grows. This makes the proposed method more efficient compared to previous methods.
- The paper presents good experimental evaluations and comparisons with previous methods.
- The paper also presents experimental evidence to study the benefits of each proposed component in ablation studies.

**Weaknesses:**

-Table 2 doesn't mention AP for small objects. The proposed method performs better than previous methods on medium and large objects. How does this compare to small objects?

-How do the ablation studies change on the COCOHumanParts dataset? I am curious about the performance of different components of the proposed method on different object sizes: small, medium, and large.

**Questions:**

Please see the Weakness section.

---

> ### Author Response · Authors · 2023-11-14
>
> Dear Reviewer NDUe,
>
> Thank you for your constructive feedback and insightful questions regarding our paper. We appreciate the opportunity to clarify the aspects you have highlighted. Please find our responses to your queries below.
>
> 1. AP for Small Objects:
>
> A: We acknowledge your observation about the absence of AP for small objects in Table 2. It is important to note that in the original Hier R-CNN paper, small objects are further categorized into small and tiny objects, and AP for small objects is not explicitly provided. Similarly, another comparison method, BPJDet, also does not offer AP details for small objects. We will include the APs for small objects in our revised paper. We emphasize that Table 2 demonstrates our method's competitive object detection accuracy relative to state-of-the-art approaches. However, the critical metric for our task is association accuracy, where, as shown in Table 3, our method achieves superior performance. This distinction underscores our method's effectiveness in part-body association, which is central to our research objective.
>
> 2. Ablation Study on COCOHumanParts Dataset:
>
> A: We have not conducted ablation studies on the COCOHumanParts dataset in our original submission. We acknowledge the importance of such analysis for a comprehensive understanding of our method's performance across different object sizes (small, medium, and large). We would like to highlight that our ablation studies were conducted on the BodyHands dataset, a large-scale dataset with unconstrained images and annotations for hand and body locations and correspondences. This dataset comprises 18,861 training images and 1,629 test images. The results from the BodyHands dataset should provide convincing evidence of our method's efficacy. Nevertheless, we understand the value of additional ablation studies on the COCOHumanParts dataset and, time permitting, we aim to conduct this analysis and share the findings.
>
> Your feedback has been invaluable in helping us improve the clarity and comprehensiveness of our research. We are committed to addressing these points and enhancing the overall quality of our paper.

---

> > ### Comment · Reviewer_NDUe · 2023-11-22
> >
> > I thank the authors for addressing my concerns. I have read the other reviews, and I would like to keep my original score.

---

### Official Review · Reviewer_AkLB · 2023-10-31

**Soundness:** 4 excellent
**Presentation:** 3 good
**Contribution:** 3 good
**Rating:** 8
**Confidence:** 3

**Summary:**

The paper presents a one-stage anchor-free approach (PBADet) for part-body association detection via singular part-to-body center offset that captures the relationship between parts and their parent bodies. It is an extension to one-stage anchor-free object detection methods for part-body association to identify part-body relationships. The approach supports multiple parts-to-body associations without compromising the accuracy and robustness of the detection process. The approach is evaluated on BodyHands, COCOHumanParts, and CrowdHuman datasets and the performance is comparable to the state-of-the-art (SOTA).

**Strengths:**

The paper is written well and easy to follow. The idea is very good and is inspired by the recent task alignment learning to enhance interaction between the two tasks for high-quality predictions.

The rationale behind the anchor-free prediction, and part-to-body associations using models like YOLOv5, YOLOv7, and YOLOV8 is justified.

A thorough experimental evaluation using well-known benchmarked datasets. On each dataset, the performance of the proposed approach is compared to the state-of-the-art and explains the performance gain and its impact on the overall accuracy.

The importance of the individual module model is experimentally evaluated.

Interesting visualization to show qualitative comparison and highlight the erroneous predictions.

**Weaknesses:**

The anchor-free object representation uses multi-scale feature maps. How many scales have been used?  Is this the same as the backbone (e.g., YOLOv7) feature representation?

The approach uses a singular part-to-body center offset. The body center is the bounding box center or something else. Also, how important is accuracy in detecting the body center influencing the overall performance?

It would be nice to have a section on model capacity and computational complexity (e.g. Params, GFLOPS, per-image inference time, etc) to further improve the article. This should be compared to the other SOTA models.

The multi-scale module has a significant impact on the performance improvement w.r.t. the baseline. What could be the reason?

The loss weights ($\lambda$ and NMS thresholds $\tau$ values) are optimal for a given dataset?

**Questions:**

Please refer to the "Weakness" section.

---

> ### Author Response · Authors · 2023-11-14
>
> Dear Reviewer AkLB,
>
> We greatly appreciate your thorough review and constructive feedback on our paper. Your points have helped us identify areas for clarification and improvement. Please find our responses to your queries below.
> 1. Number of Scales Used:
>
> A: In our experiments, the multi-scale feature maps align with the backbone feature representations. Specifically, the YOLOv7 model generates P3-P5 outputs (i.e., three scales), and the YOLOv5l6 model provides P3-P6 outputs (i.e., four scales). This consistency with the backbone ensures optimal utilization of the multi-scale feature capabilities.
>
> 2.  Clarification on Body Center / Part-to-Body Center Offset:
>
> A: In our model, the body center is determined as the center of the bounding box, following standard practices in bounding box detection. Specifically, the center is calculated using the formula ((r+l)/2, (b+t)/2) for a bounding box defined by coordinates (r, l, t, b). Our ablation study (Table 4) demonstrates that incorporating association loss (for detecting part-body center offset) does not adversely affect hand detection accuracy, showcasing the robustness of our approach.
>
> 3. Model Capacity and Computational Complexity:
>
> A: We thank you for highlighting the importance of discussing model capacity and computational complexity. Our paper provides detailed information on model parameter sizes and input image sizes in Tables 1 and 3. To further emphasize the efficiency of our model, we have included a comprehensive analysis of inference speed.
>
> Specifically, we have assessed the frames per second (FPS) of PBADet using YOLOv7 and YOLOv5l6 detectors on a GPU V100. With an input size of 1280x1280, PBADet achieves approximately 84 FPS with YOLOv7 and 63 FPS with YOLOv5l6. These numbers indicate that PBADet is well-suited for real-time applications, demonstrating significant efficiency in processing.
>
> For context, we compared this with the ResNet-101-FPN-based detector Mask R-CNN, which is utilized in methods such as BodyHands [1] and Hier R-CNN [2]. The official FPS for Mask R-CNN is about 10 to 20 per image on an NVIDIA V100 GPU & NVLink [3] [4]. When considering the additional computational load due to the association network in BodyHands [1] and Hier R-CNN [2], it is evident that these methods lag significantly behind PBADet in terms of speed. Note that our method has an input size of 1024x1024 while BodyHands [1] and Hier R-CNN [2] use an input size of 1536x1536 (note that the original Mask-CNN uses 800x800), which further enlarges the FPS gap. This disparity in processing efficiency highlights the advanced capability of PBADet, particularly when it comes to real-time processing demands.
>
> [1] Whose Hands Are These? Hand Detection and Hand-Body Association in the Wild, CVPR 2022
>
> [2] Hier R-CNN: Instance-Level Human Parts Detection and A New Benchmark, TIP 2020
>
> [3] Mask R-CNN
>
> [4] https://github.com/facebookresearch/detectron2/blob/main/MODEL_ZOO.md
>
> 4. Impact of Multi-Scale Module:
>
> A: Our ablation study in Table 4 indicates that interpreting the center offset as a normalized absolute distance without multi-scale considerations leads to a reduction in hand prediction accuracy and joint AP. This is because each point in the multi-scale dense feature predicts a center offset, and it is challenging to achieve convergence for different scales to output the same absolute distance. The multi-scale module thus plays a critical role in enhancing performance.
>
> 5. Optimization of Loss Weights and NMS Thresholds:
>
> A: The loss weights during training remain the same across different datasets. However, the NMS thresholds are not loss weights and are post-processing parameters in object detection. NMS thresholds are optimally selected for each dataset since different datasets present varying levels of crowdedness. This approach is in line with protocols followed in previous methods, such as BPJDet, ensuring a fair and effective comparison.
>
> We hope these clarifications address your concerns and contribute to a better understanding of our work. We are committed to making the necessary revisions in our paper to reflect these clarifications and enhance its overall quality.

---

> > ### Comment · Reviewer_AkLB · 2023-11-20
> >
> > Thanks for addressing my concerns.
> >
> > I have gone through the reviews of other reviewers comments/questions and authors' reply to those. Most of raised questions are addressed convincingly. Thus, I am inclined in accepting. Thank you.

---

### Official Review · Reviewer_BLsk · 2023-10-31

**Soundness:** 2 fair
**Presentation:** 3 good
**Contribution:** 3 good
**Rating:** 6
**Confidence:** 4

**Summary:**

-	It would be better if more details about the association process could be given.
-	How to choose the value of the hyper-parameter K used in L_{assoc}? The experiments would be more comprehensive if the value of K could be analyzed in the ablation study.
-	The reviewer wonders whether the models are trained from scratch or the pretrained YOLO weights are used. It is not mentioned in the paper.

**Strengths:**

-	Compared to the body-to-part center offset used in the previous one-stage method BPJDet, the part-to-body center offset proposed in this paper has better scalability in terms of the number of parts and avoids degrading the overall object detection performance.
-	The PBADet method proposed in the paper achieves state-of-the-art performance on BodyHands, COCOHumanParts and CrowdHuman datasets.
-	This paper is well organized, clearly presented, and effectively clarifies the differences from previous methods.

**Weaknesses:**

-	The author states that the proposed part-to-body center offset guarantees a one-to-one correspondence between parts and bodies. However, the paper does not elaborate on how the situation is handled when the predicted center offsets of multiple parts with the same category point to the same body. How to define the order for associating multiple parts with the same category? These details are not described in Section 3.4.
-	In Figure 2, it seems that 'P3' and 'P5' in the 'Multi-scale features' are labeled incorrectly.

**Questions:**

-	It would be better if more details about the association process could be given.
-	How to choose the value of the hyper-parameter K used in L_{assoc}? The experiments would be more comprehensive if the value of K could be analyzed in the ablation study.
-	The reviewer wonders whether the models are trained from scratch or the pretrained YOLO weights are used. It is not mentioned in the paper.

---

> ### Author Response · Authors · 2023-11-14
>
> Dear Reviewer BLsk,
>
> Thank you for your valuable feedback and insightful comments on our paper. We appreciate the opportunity to clarify and address your concerns.
>
> 1.  Handling Multiple Parts Pointing to the Same Body:
>
> A: We acknowledge your concern regarding how our method handles situations where multiple parts of the same category point to the same body. As detailed in Section 3.4 "DECODING PART-TO-BODY ASSOCIATIONS", our network outputs the center offset of a part, pointing to the body to which the part belongs. We use Euclidean distance between the estimated body center (part center plus the center offset) and the centers of unassigned bodies that enclose the part. The body with the smallest distance (i.e., the distance of the estimated body center and body center) is then assigned as the corresponding body for the given part. This approach ensures accurate and efficient part-to-body association.
>
> 2. Typographical Error in Figure 2:
>
> A: We are grateful for your observation regarding the labeling error in Figure 2, where 'P3' and 'P5' in the 'Multi-scale features' were incorrectly labeled. We have corrected this typo in the revised version of the paper to ensure clarity and accuracy in our visual representations.
>
> 3. Clarification on the Hyper-parameter K in L_{assoc}:
>
> A: Regarding the hyper-parameter K used in L_{assoc}, we have chosen K=13, aligning with the default value used in established anchor-free object detection methods such as YOLOv7 and YOLOv8. This choice is substantiated in our ablation study (Table 4), where it is shown that performance degrades when K equals the number of anchor points (i.e., w/o Task-Align).
>
> 4. Model Training from Scratch or Pretrained YOLO weights:
>
> A: We appreciate your inquiry about whether our model was trained from scratch or using pretrained YOLO weights. To clarify, we followed the same protocol established in the BPJDet paper [1] that uses pretrained YOLO weights. This implementation detail will be explicitly stated in the revised manuscript to avoid any ambiguity.
>
> [1] Body-Part Joint Detection and Association via Extended Object Representation
>
> Your feedback has been instrumental in enhancing the clarity and accuracy of our work. We are committed to making the necessary revisions to address these points thoroughly.

---

### Author Response · Authors · 2023-11-22
**Submission of Revised Paper**

Dear Reviewers,

We are writing to inform you that we have submitted the revised version of our paper. We would like to express our sincere gratitude for your insightful and constructive comments and feedback. Your expert critiques have been invaluable in guiding the improvements we have made to our paper.

In response to the points raised during the review process, we have made the following comprehensive revisions:

- **Related Work Section**: We have expanded the discussions in the related work section to more clearly delineate the distinctions and parallels between our work and the key references cited.
- **Experimental Analyses**: Additional experimental analyses have been included in the Appendix, specifically in sections A.2 (Quantitative Comparison to Multi-person Pose Estimation), A.3 (Experiments on Animal Datasets), A.4 (More Ablation Study - Anchor-based Comparison), and A.5 (Detailed Results of COCOHumanParts of small objects). These analyses provide enriched data and reinforce the conclusions of our study.
- **Figures Revision**: We have revised Figures 1, 2, 3, and 4, addressing specific issues such as correcting typos and changing arrows to improve their clarity and the conveyance of information.
- **Implementation Details**: In Section 4.1 (Evaluation Protocols), we have added details about using pretrained weights, enhancing the clarity and reproducibility of our methodology.

We believe that these comprehensive revisions have notably elevated the quality, clarity, and robustness of our research. We are hopeful that the paper now aligns more closely with the esteemed standards of the conference. We are grateful for the opportunity to refine our work and appreciate the dedication and effort you have invested in reviewing our paper.

Thank you once again for your invaluable assistance in enhancing the quality of our work. We look forward to your continued guidance and are hopeful for favorable consideration of our paper.

---

### Meta-Review · Area_Chair_W5fD · 2023-12-11

**Metareview:**

This work received six pieces of review comments, and five of them tend to accept this paper. Reviewers mentioned that the part-to-body center offset scheme proposed in this work has scalability in terms of the number of parts, which also avoids degrading the overall object detection performance. Overall, the idea is good and the results are promising. The main concern of the reviewer who recommends borderline rejection is the similarity of this work w.r.t a TMM 2022 work -  Grouping by Center: Predicting Centripetal Offsets for the Bottom-up Human Pose Estimation. TMM, 2022. The authors clarified the differences between this work and the TMM work, in terms of problems and challenges. Considering that the method is useful for the part-body association problem, the AC recommends accept. Authors need to carefully improve the work and address reviewers' comments in the camera ready version.

**Justification For Why Not Higher Score:**

There are some similarities of this method w.r.t an existing pose estimation work, though the tasks are different.

**Justification For Why Not Lower Score:**

The method is good and useful for body part association.

---

### Decision · Program_Chairs · 2024-01-16

Accept (poster)